

# Effects of nanoscale zinc oxide treatment on growth, rhizosphere microbiota, and metabolism of *Aconitum carmichaelii*

Cun Chen[1,2], Yu-yang Zhao[1], Duo Wang[3], Ying-hong Ren[2], Hong-ling Liu[2], Ye Tian[4], Yue-fei Geng[4], Ying-rui Tang[2] and Xing-fu Chen[1]

[1] College of Agronomy, Sichuan Agricultural University, Chengdu, Sichuan, China
[2] College of Chemistry and Life Science, Sichuan Provincial Key Laboratory for Development and Utilization of Characteristic Horticultural Biological Resources, Chengdu Normal University, Chengdu, Sichuan, China
[3] Key Laboratory of Bio-Resource and Eco-Environment of Ministry of Education, College of Life Sciences, Sichuan University, Chengdu, Sichuan, China
[4] Sichuan Jianengda Panxi Pharmaceutical Co. LTD, Xichang, Sichuan, China

## ABSTRACT

Trace elements play a crucial role in the growth and bioactive substance content of medicinal plants, but their utilization efficiency in soil is often low. In this study, soil and *Aconitum carmichaelii* samples were collected and measured from 22 different locations, followed by an analysis of the relationship between trace elements and the yield and alkaloid content of the plants. The results indicated a significant positive correlation between zinc, trace elements in the soil, and the yield and alkaloid content of *A. carmichaelii*. Subsequent treatment of *A. carmichaelii* with both bulk zinc oxide (ZnO) and zinc oxide nanoparticles (ZnO NPs) demonstrated that the use of ZnO NPs significantly enhanced plant growth and monoester-type alkaloid content. To elucidate the underlying mechanisms responsible for these effects, metabolomic analysis was performed, resulting in the identification of 38 differentially expressed metabolites in eight metabolic pathways between the two treatments. Additionally, significant differences were observed in the rhizosphere bacterial communities, with *Bacteroidota* and *Actinobacteriota* identified as valuable biomarkers for ZnO NP treatment. Co-variation analysis further revealed significant correlations between specific microbial communities and metabolite expression levels. These findings provide compelling evidence that nanoscale zinc exhibits much higher utilization efficiency compared to traditional zinc fertilizer.

## INTRODUCTION

Herbal medicines, being natural, are favored over synthetic remedies by a significant portion of the global population. There is a growing interest in the health and wellness benefits of herbs and botanicals. *Aconitum*, a large genus of the Ranunculaceae family, has been of interest since ancient times due to its diterpene alkaloids, which range from relatively harmless to fatally poisonous (*Kuniyal, Bhadula & Prasad, 2002*; *Singhuber et al., 2009*). *A. carmichaelii* is a traditional Chinese medicinal herb, that has been clinically

Corresponding author
Xing-fu Chen,
chenxingfu@sicau.edu.cn

used for almost two thousand years. Its lateral roots, known as "fuzi" in China, "bushi" in Japan, and "kyeong-po buja" in Korea, have been extensively used as a cardiotonic, anti-inflammatory, analgesic, and diuretic agent for the treatment of various ailments, including colds, diarrhea, heart failure, polyarthralgia, and edema (*Xiao et al., 2006*; *Zhou et al., 2014*). The primary active ingredients of *A. carmichaelii* are the diester-type alkaloids, including aconitine, hypaconitine, and mesaconitine, and the monoester-type alkaloids, including benzoylaconine, benzoylhypaconine, and benzoylmesaconine (*Wang et al., 2009*; *Nesterova et al., 2014*). Although diester-type alkaloids have the highest concentration and strongest effect, they are also known to be highly toxic. In contrast, monoester-type alkaloids exhibit only 1/100 to 1/200 of the toxicity levels to tissue cells (*He et al., 2023*). However, during the cultivation of *A. carmichaelii*, a common problem is the low yield of lateral roots and low content of monoester-type alkaloids.

In the cultivation of medicinal plants, micro-fertilizers are commonly used to enhance both yield and bioactive substance content. Trace elements are essential nutrients for plant growth, and although the demand for them is relatively small, each trace element plays an important biological role (*Parshuram, Himangshu & Ranjan, 2022*; *Xiaodong et al., 2022*). It has been reported that the application of micro-fertilizer (0.2% B, 0.3% Cu, 0.8% Fe, 0.4% Mn, 0.3% Zn) can increase the umbel number, seed yield per plant and per decare, thousand seed weight, as well as the essential oil content of anise (*Pimpinella anisum*) (*Günay & Gümüşçü, 2023*). The application of copper fertilizer has been found to contribute to the accumulation of tanning substances, ascorbic acid, and carotene in the harvested herb material of *Echinacea purpurea* (*Zharkova, Sukhotskaya & Ermokhin, 2020*). Boron plays a significant role in various physiological processes of plants. It is involved in nucleic acid synthesis, phenolic metabolism, carbohydrate biosynthesis and translocation, pollen tube growth, and root elongation. Copper acts as a potent activator of enzymes and plays a crucial role in facilitating the absorption of nitrogen compounds, indirectly leading to enhanced chlorophyll production and increased sugar content. Iron is vital for promoting chlorophyll formation and facilitating the functioning of enzymes involved in various biological processes, including cellular respiration, cell division, and growth. Manganese and zinc exert significant influence on protein biosynthesis by modulating peptidase activity and regulating protein metabolism. Widely recognized as a critical element, zinc serves as a versatile component of enzymes, acting as a metal cofactor or fulfilling functional, structural, and regulatory roles in numerous enzymatic reactions (*Hänsch & Mendel, 2009*; *Lewis, 2019*; *Singh & Dwivedi, 2019*; *Jaiswal et al., 2022*). However, the nutrient utilization efficiency of conventional micronutrient fertilizers is almost as low as 30 to 40% (*Van Dijk & Meijerink, 2014*; *Alzreejawi & Al-Juthery, 2020*). Existing research has found that reducing fertilizer particle size can increase the quality surface area ratio of particles. When the particle size is reduced to the nanometer level, a significant amount of nutrient ions can be slowly and stably absorbed over a long period of time, ensuring the nutritional balance of crops throughout the growth period and ultimately improving agricultural production (*Subramanian et al., 2015*; *Monreal et al., 2016*; *Ndaba et al., 2022*). Due to their excellent performance, nanometer particles have gradually emerged as important in agricultural production, and metal oxide nanoparticles (MeO NPs) are among the most

relevant types of nanoparticles (*Buzea, Pacheco & Robbie, 2007*; *Martínez-Fernández et al., 2017*; *Zuverza-Mena et al., 2017*).

In recent years, MeO NPs have sparked significant interest in promoting plant growth and yield, making them increasingly applicable in agriculture as nano-fertilizers, growth promoters, nano-pesticides, soil amendments, and so on (*Fraceto et al., 2016*; *Wang et al., 2016*). The application of $TiO_2$ and $SiO_2$ NPs as elicitors has been demonstrated to effectively elevate parthenolide levels in Tanacetum parthenium by influencing gene expression and translation levels, thereby impacting metabolite production (*Khajavi, Rahaie & Ebrahimi, 2019*). Similarly, investigations on medicinal plants have shown that the utilization of ZnO and $Fe_2O_3$ NPs can enhance the underground biomass and root diameter of *Salvia miltiorrhiza*, which is believed to be due to the alteration of rhizosphere microorganisms induced by MeO NPs (*Wei et al., 2021*). Furthermore, *Wang et al. (2022)* found that $Fe_3O_4$ NPs have the ability to promote the growth of *Dendrobium huoshanense*, induce early flowering, increase sugar content and photosynthesis in the plant, and elevate MDA levels and antioxidative enzyme activity. They attributed these effects to the modification of related metabolic pathways in *Dendrobium huoshanense* caused by $Fe_3O_4$ NPs, resulting in a significant increase in the concentration of primary active substances, including polysaccharides, phenols, flavonoids, and anthocyanins. As the demand for NPs, including in agriculture, continues to rise, understanding the effects and mechanisms of MeO NPs on plant growth is crucial. The identification of the type of micro-fertilizers that significantly improves the yield and content of monoester-type alkaloids in *A. carmichaelii* and whether its nanoscale form has greater potential is of interest to us.

This study investigates the relationship between soil chemical properties and the yield/monoester-type alkaloids of *A. carmichaelii* in various planting areas of Butuo County, Xichang City, Sichuan Province. The primary objective is to identify the trace elements that have the greatest impact on the plant's yield/monoester-type alkaloids content. Based on the results of the above investigation, zinc was selected as the trace element that has the greatest impact on *A. carmichaelii* growth. Further tests were conducted using nanoscale zinc oxide (ZnO NPs) and bulk ZnO to study the effects of ZnO NPs on *A. carmichaelii* growth and metabolism, as well as their impact on rhizosphere bacteria. In addition, we aimed to establish a correlation between alterations in metabolites and microorganisms to elucidate the mechanisms through which ZnO NPs affect *A. carmichaelii* growth and metabolism, while identifying key factors that influence plant growth and development under the influence of ZnO NPs.

## MATERIALS AND METHODS

### Sampling investigation of soil trace elements' correlation with yield and monoester-type alkaloids in *A. carmichaelii*

The sampling area, Butuo County, is located in Xichang City and covers an area of 1685 $km^2$, situated between the north latitudes of 27°16′ to 27°56′ and east longitudes of 102°43′ to 103°04′ in southwestern Sichuan Province, China. The main herb and economic crop in this region is *A. carmichaelii*, which is exported to East Asian countries including

Japan, Korea, Mongolia, and India (*Yu et al., 2016*).The maps of the sample locations were constructed using ArcMap Desktop (version 10.8) with the Spatial Analyst module (*Liu, Shao & Wang, 2013*; *Mahapatra et al., 2020*).

Soil and plant samples were collected from 22 locations in six different villages within Butuo (Fig. S1). The sample collection process involved employing a five-point sampling method at each location. A total of 10 plants were sampled from each plot. Soil samples were collected from the cultivated layer within a 20 cm depth surrounding the roots of the plants at each sampling point, resulting in approximately 0.5 kg of mixed soil sample. All soil samples underwent a natural air-drying process, followed by grinding and sieving through a two mm sieve. These processed soil samples were then utilized for soil parameter analysis in the laboratory.

### Soil trace element analysis

Available zinc (AZn) was estimated by triplicate soil extractions with 0.02 M $Na_2$-EDTA + 0.5 M $NH_4Ac$ at pH 4.65, using 5 mL solution and shaking for 30 min (*Campillo-Cora et al., 2019*). Available manganese (AMn), available boron (AB), available iron (AFe) and available copper (ACu)were determined by shaking 5 g of soil with 10 mL diethylene triamine penta acetic acid (DTPA) extractant (pH 7.3) for 2 h (*Lindsay, 1978*). Total trace element assays used microwave-assisted digestion with some modifications. The included elements were total zinc (TZn), total manganese (TMn), total boron (TB), total iron (TFe) and total copper (TCu) (*Bettinelli et al., 2000*). For trace element assessment, 0.3 g of each soil sample was digested with a mixture of 5 mL nitric acid (67% w/v), 3 mL hydrochloric acid (37% w/v), and 1 mL hydrofluoric acid (40% w/v) in a microwave chemical reactor (MDS-6G, SINEO, Shanghai, China) with a temperature program of 160 °C for 10 min, then increased to 210 °C for 30 min. The resulting clear brownish solution was treated with an acid scavenger (TK12, SINEO, Shanghai, China) for 30 min and diluted to 50 ml with 2% nitric acid. All samples were centrifuged at $8,586 \times g$ for 10 min and the supernatant was filtered through a 0.45 µm membrane filter. The resulting samples were analyzed for heavy metals using an atomic absorption spectrophotometer (AA-7020, EWAI, Beijing, China). Three replicates were prepared for each analytical sample.

### Analysis of yield and monoester-type alkaloids

The yield was obtained by measuring the fresh weight of lateral roots from 10 plants at each sampling point, after removing the soil. To analyze the content of monoester-type alkaloids, lateral root powder was sieved through a sieve with a diameter of 0.15 mm. Next, 1 g of the powder was added to 10 mL of 1% hydrochloric acid, and the resulting mixture was thoroughly shaken and sonicated for 30 min at 150 W and 40 kHz. After allowing the mixture to stand at room temperature for 30 min, it was centrifuged at $8,586 \times g$ for 10 min. The supernatant was collected and filtered using a 0.45 µm membrane filter. Subsequently, 20 µL of the filtered solution was taken for further analysis using high performance liquid chromatography(HPLC). In the HPLC method, a C18 column (250 × 4.5 mm, 5 µm) was used at 25 °C. A 20 µL sample was injected and monitored at 235 nm. A mobile phase of acetonitrile-tetrahydrofuran (9:1):0.1 mol/L ammonium acetate

solution (70:30 v/v) with a flow rate of 0.8 mL min −1 was utilized. The analyses were performed using a Shimadzu LC-20A liquid chromatograph with a diode array detector (CBM20A, Shimadzu, Tokyo, Japan).

## ZnO NPs-treated pot experiments

The test was conducted in a greenhouse located at Chengdu Normal University (30°40′47″N, 103°49′16″E) in Chengdu, Sichuan Province, China. Pots were prepared and filled with 6 kg of soil for the plant growth experiment. The soil used in the study was obtained from a farm near the university, and natural drying was used to remove inclusions such as rocks and plant matter. The dried soil samples were ground to pass through a two mm sieve and mixed thoroughly and homogeneously. The soil had a pH of 6.4, electrical conductivity (EC) of 143.1 ms/m, soil organic matter (SOM) of 161.7 mg/kg, available nitrogen (AN) of 75.62 mg/kg, available phosphorus (AP) of 26.53 mg/kg, available potassium(AK) of 110.32 mg/kg, and available zinc (AZn) of 0.35 mg/kg.

The *A. carmichaelii* plants used in this study were artificially cultivated in Butuo County. ZnO NPs (with a diameter of $50 \pm 10$ nm and 99.9% purity) were purchased from Shanghai Macklin Biochemical Co. Ltd and used as test NPs in this study. In each pot, we applied 100 mL of a solution consisting of either distilled water (CK), bulk ZnO at a concentration of 30 mg/L, or ZnO NPs at the same concentration. To avoid particle aggregation, both bulk ZnO and ZnO NPs were suspended directly in deionized water, followed by ultrasonic vibration (200 W, 40 kHz) for 30 min prior to use. These treatments were applied three times to each pot.

## Metabolite extraction

The metabolites of *A. carmichaelii* lateral roots were analyzed soon after harvest. The samples were freeze-dried, ground into powder, and dissolved in one mL of extract solution (methanol: water = 7: 3, v/v) that had been pre-cooled to −20 °C before being ground again. The mixtures were left overnight and then centrifuged at $13,000 \times g$ for 10 min at 4 °C. Following centrifugation, 800 μL of supernatant was collected and filtered through a 0.22 μm membrane filter before being transferred to a sample bottle for LC-MS analysis.

LC-MS/MS technology was used for conducting a broad-spectrum targeted metabolomic analysis to quantitatively measure plant metabolites. The QTRAP 6500 Plus high sensitivity mass spectrometer (SCIEX, Framingham, MA, USA) was employed for quantitative detection of MRM (Multiple reaction monitoring) in the samples. The Waters ACQUITY UPLC I-Class (Waters, Milford, MA, USA), tandem QTRAP6500 Plus high sensitivity mass spectrometer (SCIEX, Framingham, MA, USA) was utilized for effective separation and accurate quantitative detection of metabolites. An ACQUITY UPLC HSS T3 column ($100 \times 2.1$ mm, 1.8 μm, Waters) was used as the chromatographic column. The mobile phase consisted of liquid A, an aqueous solution containing 0.1% formic acid, and liquid B, 100% acetonitrile containing 0.1% formic acid. The elution process was carried out in four steps: 0–2 min, 5% B solution; 2–22 min, 5%–95% B liquid; 22–27 min, 95% B liquid; 27–30 min, 5% B solution. The flow rate used was 0.3 mL/min at a column temperature of

40 °C. For the QTRAP 6500 Plus equipped with the EST Turbo ion spray interface, the ion source parameters were set as follows: Ion source temperature: 500 °C; Ion spray voltage (IS): 4500 V (positive mode) or −4500 V (negative mode); The ion source gases I (GS1), II (2), and curtain gas (CUR) were set at 40 psi each.

## Metabolite data analysis

To identify and quantitatively analyze the metabolites, Skyline (MacCoss Lab, Seattle, WA, USA) and BGI-Wide Target-Library (BGI Co., Ltd., Shenzen, China) were used (*Wen et al., 2017*). For data statistical analysis, metabolite classification annotation, and functional annotation, MetaboAnalyst 5.0 was employed (*Pang et al., 2022*).

The variable importance in projection (VIP) of the orthogonal partial least-squares discriminant analysis (OPLS-DA) model was used to screen out differential metabolites between the control group and the experimental group (*Jia et al., 2021*; *Zhou et al., 2021*). Metabolites were considered differential if they had VIP ≥1, absolute Log2FC (fold change) ≥1, and fold change (FC) ≥2 or FC ≤0.5 for group discrimination (*Cao et al., 2019*). To confirm the OPLS-DA model's screening accuracy, principle component analysis (PCA) was used to assess the overall clustering pattern of the two treatment groups. The correlation between samples was calculated using Pearson correlation.

## Rhizosphere microbial analysis

DNA samples for the soil microbiome were collected from the rhizosphere area of the plants at harvest. All high-throughput sequencing was completed by Biomarker Technologies Co., Ltd (Beijing, China). DNA purification was performed using the Soil DNA Kit (MN NucleoSpin 96 Soil). The method adhered to the manufacturer's instructions for the extraction procedure. Initially, the soil was weighed and 700 μL of Buffer SL1 and Buffer SL2 was added. The MN Bead Tube was then filled with fresh lysis buffer up to the 1.5 mL mark, followed by the addition of 150 μL of Enhancer SX. To prevent leakage, the cap was tightly closed. Subsequently, the sample was lysed by horizontally attaching the MN Bead Tube to a vortexer and vortexing at full speed and room temperature (25 °C) for 5 min. Afterward, the tube underwent centrifugation at 11,000 × *g* for 2 min to eliminate any foam generated by the detergent. Following this step, 150 μL of Buffer SL3 was added, and the mixture was vortexed for 5 s. An incubation period of 5 min at 0−4 °C ensued. The tube was then centrifuged at 11,000 × *g* for 1 min to separate the lysate. To remove any impurities, the lysate was filtered through an appropriate filter. The DNA was bound to a silica membrane according to the manufacturer's instructions. Subsequently, the silica membrane was washed and dried following the kit's protocol to eliminate any contaminants. To ensure optimal DNA recovery, the silica membrane was thoroughly dried. Finally, the DNA was eluted from the silica membrane using the recommended elution buffer or water, as specified in the kit's instructions. The V3-V4 hypervariable region of the 16S rRNA gene was amplified for each sample using barcoded universal primers 338 F (5′-ACTCCTACGGGAGGCAGCA-3′) and 806 R (5′- GGACTACHVGGGTWTCTAAT-3′) (*Zhang et al., 2023*). For amplicon sequencing, the Illumina NovaSeq 6000 PE250 platform (Illumina, San Diego, CA, USA) was utilized. The reads were merged and filtered by

size and quality, and then clustered into operational taxonomic units (OTUs) using an open reference strategy based on 97% identity with UPARSE (version 7.0). The bacterial 16S rRNA data were analyzed on the BMKCloud (http://www.biocloud.net) platform. Alpha-diversity was evaluated using the Shannon and Simpson index, while Beta-diversity was assessed using Principal Coordinates Analysis (PCoA) based on Bray-Curtis distance.

# RESULTS

## Correlation between soil trace elements, yield and monoester-type alkaloids content in *A. carmichaelii*

The relationship between soil trace elements, yield and monoester-type alkaloids was analyzed using Pearson correlation (Fig. 1, $p < 0.05$). The results s indicate a moderate positive correlation between the content of AMn in the soil and yield ($0.3 \leq r \leq 0.5$). Additionally, AZn is strongly positively correlated with yield and content of monoester alkaloids ($0.5 < r < 1$). On the other hand, the content of AFe and TFe exhibit a moderate negative correlation with yield ($-0.5 \leq r \leq -0.3$), while TZn content is significantly negatively correlated with the content of monoester alkaloids ($-0.5 \leq r \leq -0.3$). No significant correlation was found between the other elements and the plant. In terms of the relationship between different trace elements in soil, it was found that AZn is positively correlated with ACu ($0.5 < r < 1$) but strongly negatively correlated with AFe ($-1 < r < -0.5$) and moderate negatively correlated with TFe ($-0.5 \leq r \leq -0.3$). Moreover, ACu exhibits a moderate negative correlation with AFe ($-0.5 \leq r \leq -0.3$), while no significant correlation was found between the other elements.

## Response of plants to ZnO NPs

To investigate the impact of ZnO NPs on the growth and pharmacodynamic components of *A. carmichaelii*, we analyzed the plant height, fresh weight of the lateral root, and content of monoester-type alkaloid. The phenotypic changes of the plant were illustrated in Figs. 2A & 2B, while the lateral roots were shown in Fig. 2C. The bar graph (Figs. 2D–2F) clearly demonstrates that the application of ZnO NPs significantly promotes both plant growth and the content of monoester-type alkaloids (*$p < 0.05$, **$p < 0.01$). Specifically, our observations revealed an increase in plant height by 23.31% when compared to the distilled water treatment, and a remarkable 63.77% increase when compared to bulk ZnO treatment (Fig. 2D). The fresh weight of the lateral roots also witnessed significant improvement with an increase of 52.69% compared to the distilled water treatment and a staggering 88.58% improvement when compared to bulk ZnO treatment (Fig. 2E). Furthermore, we found an increase in the content of monoester-type alkaloids by 20.32% when compared to the distilled water treatment and an impressive 30.48% increase when compared to bulk ZnO treatment (Fig. 2F). These findings underscore the tremendous potential of ZnO NPs in enhancing plant growth and improving pharmacodynamic components in the field.

## Impact of ZnO NPs on lateral root metabolites of *A. carmichaelii*

Further metabolomics analysis was performed on the lateral roots of *A. carmichaelii* that were treated with ZnO NPs and bulk ZnO to investigate the impact of ZnO NPs on

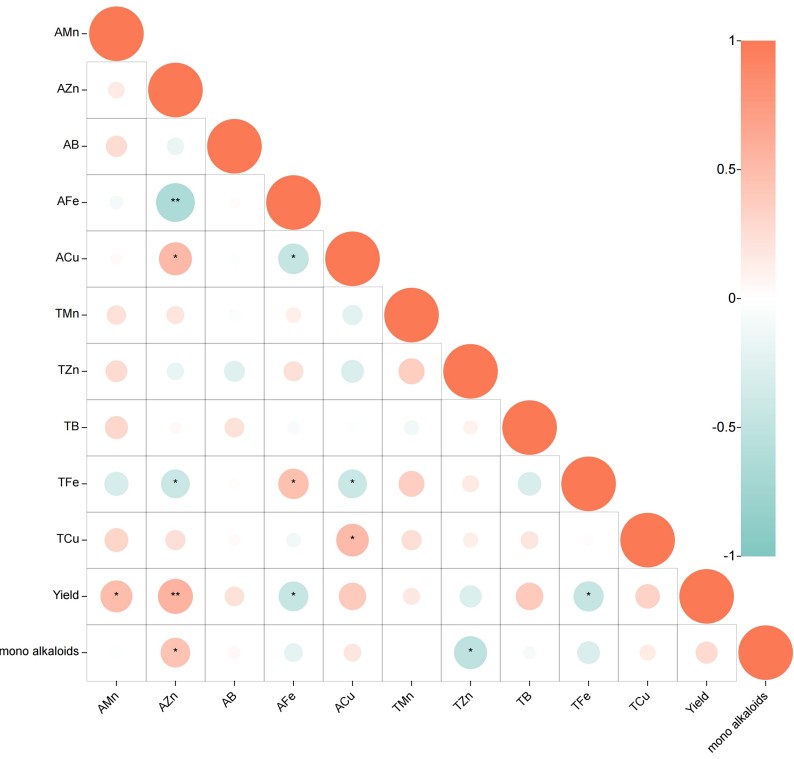

**Figure 1  Correlation between trace elements, yield and monoester-type alkaloids: Pearson correlation coefficients.** (Positive correlation in red, negative correlation in blue; $p < 0.05$, *$0.3 \leq r \leq 0.5$, or $-0.5 \leq r \leq -0.3$, **$0.5 < r < 1$, or $-1 < r < -0.5$).

the plant's metabolites. The PCA score plot showed PC1 could explain 50.7% of the total metabolic variation, while PC2 could explain 32% (Fig. 3A). Moreover, PC1 clearly distinguished the bulk ZnO and ZnO NPs treatments, revealing that the content of the selected metabolites was significantly altered between two treatments. The Volcano plot (Fig. 3B) clearly indicates a significant difference between the bulk ZnO and ZnO NPs treatments, as evidenced by the separation of the data points into two distinct clusters. The significant 38 metabolites were selected based on VIP $\geq 1$, absolute Log2FC (fold change) $\geq 1$, FC $\geq 2$ or FC $\leq 0.5$, and $P < 0.05$, with 23 metabolites upregulated and 15 metabolites downregulated (Table S1 & Fig. 3B). The differential metabolites between the two treatments encompassed a wide array of compound classes, including carbohydrates, amino acids, guanidines, amines and derivatives, polyketides, terpenoids, flavonoids, branched unsaturated hydrocarbons, hydroxyindoles, alkaloids, and more. The ZnO NPs treatment upregulated certain metabolites involved carbohydrate and amino acids related to plant growth or stress resistance, such as D-Mannosamine and D-Proline, while downregulating certain glycosides and terpenes, including Rehmannioside C. Notably, the alkaloid 14-Benzoylaconine, a clinically significant component of *A. carmichaelii*, showed an increase. The correlation of these 38 metabolites was presented in a correlation heatmap (Fig. 3C).

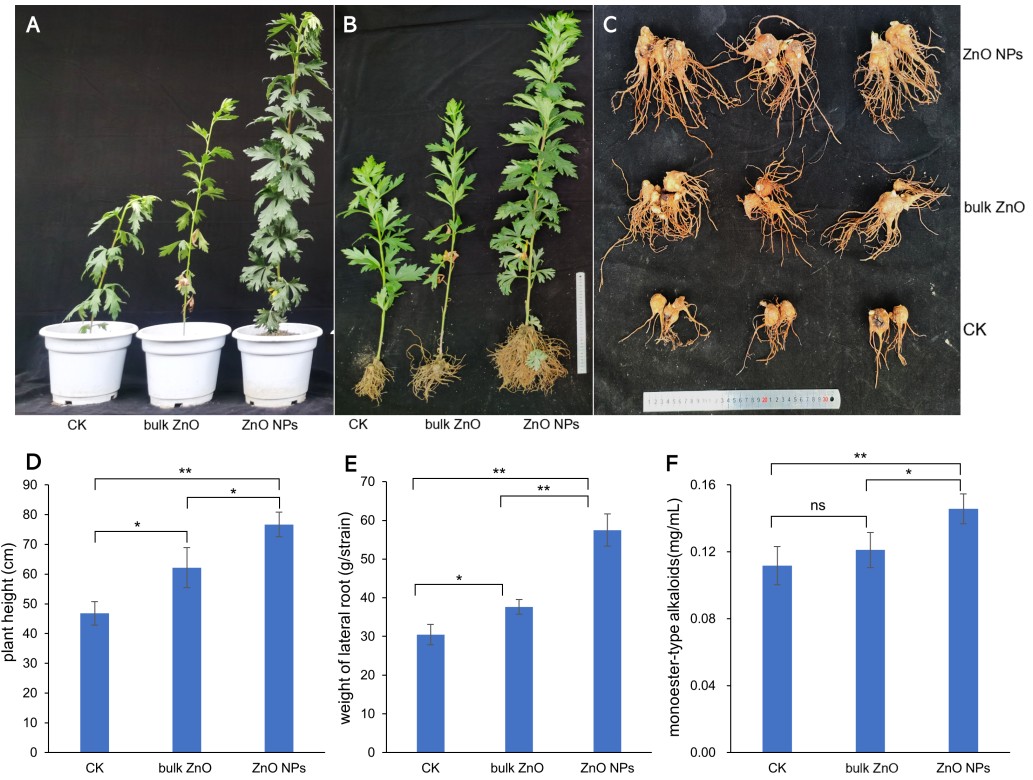

**Figure 2** **Effect of ZnO NPs on phenotypes and pharmacodynamic component of *A. carmichaelii*,**
**(\*$p < 0.05$, \*\*$p < 0.01$).** (A–B) Phenotypic changes of the plant. (C) Phenotypic changes of the lateral roots. (D) Effects of ZnO NPs on plant height. (E) Effects of ZnO NPs on fresh weight of the lateral root. (F) Effects of ZnO NPs on monoester-type alkaloids.

To gain a comprehensive understanding of how plants respond to ZnO NPs, a metabolic pathway analysis was conducted using MetaboAnalyst 5.0. This study assessed 53 pathways in the lateral roots of *A. carmichaelii* and identified eight pathways where differential metabolites are involved. The metabolic pathways were arginine and proline metabolism (1), aminoacyl-tRNA biosynthesis (2), flavone and flavonol biosynthesis (3), glutathione metabolism (4), alkaloid biosynthesis (5), ascorbate and aldarate metabolism (6), tryptophan metabolism (7), and amino sugar and nucleotide sugar metabolism (8). Among these eight metabolic pathways, five of them were significantly upregulated by ZnO, excluding flavone and flavonol biosynthesis (3), ascorbate and aldarate metabolism (6), and amino sugar and nucleotide sugar metabolism (8), as shown in Fig. 3D.

### Diversity, composition, and functional prediction of rhizosphere bacteria

Similarly, rhizospheric bacteria of *A. carmichaelii* treated with ZnO NPs and bulk ZnO were assessed to investigate the impact of ZnO NPs on the plant's rhizosphere. The analysis of the bacterial 16S rRNA data was performed on the BMKCloud (http://www.biocloud.net) platform. The *P* values of alpha-diversity index (Shannon and Simpson) in rhizosphere bacteria were all smaller than 0.05, suggesting that there was a statistically significant

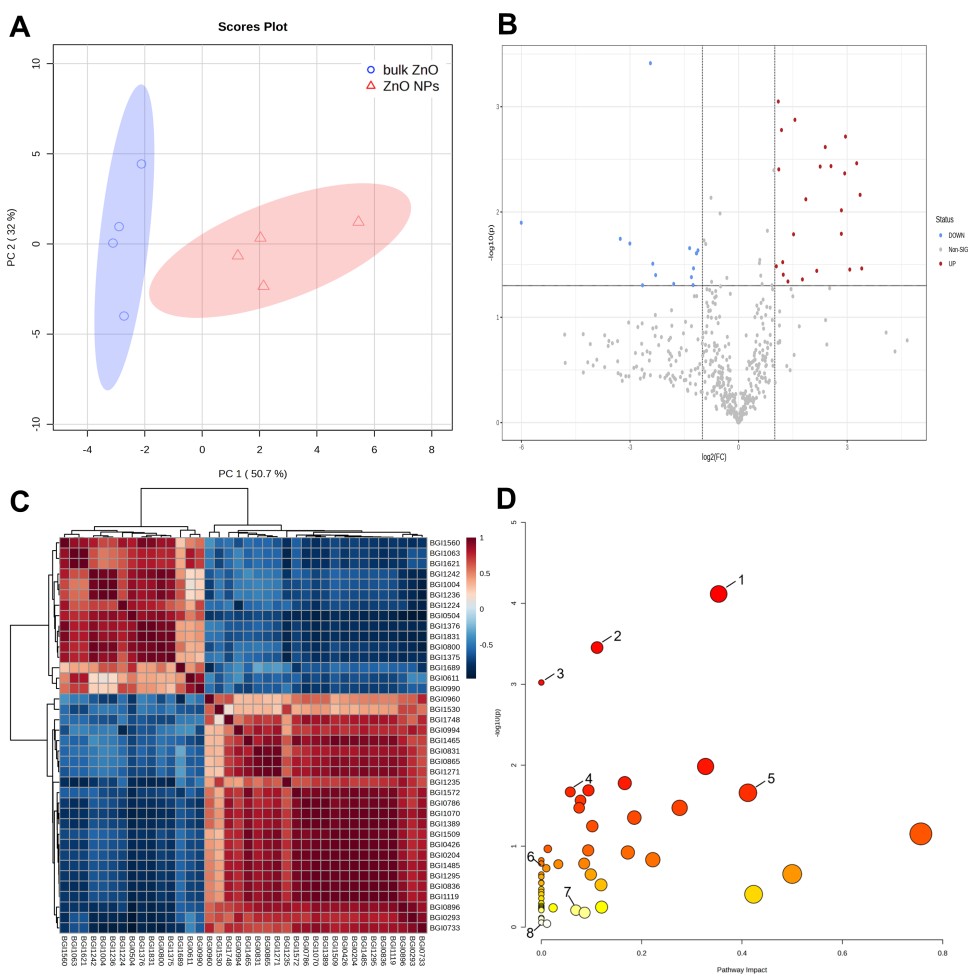

**Figure 3** Identification of metabolites differentially accumulated in *A. carmichaelii* under bulk ZnO and ZnO NPs treatments. (A) PCA score plot. (B) Volcano plot, where red dots signify upregulated metabolites and blue dots signify downregulated ones.(C) Correlation analysis of differential metabolites. (D) Pathway analysis with MetaboAnalyst 5.0. (1) Arginine and proline metabolism (2) Aminoacyl-tRNA biosynthesis (3) Flavone and flavonol biosynthesis (4) Glutathione metabolism (5) Alkaloid biosynthesis (6) Ascorbate and aldarate metabolism (7) Tryptophan metabolism (8) Amino sugar and nucleotide sugar metabolism.

difference in species richness and relative abundance between bulk ZnO and ZnO NPs treatments in rhizosphere bacteria (Figs. 4A & 4B). Beta diversity was analyzed with Principal Coordinates Analysis (PCoA), using Bray-Curtis dissimilarity. The PCoA plot revealed distinct differences between the rhizosphere soil bacterial communities from the ZnO NPs treatment and those associated with the bulk ZnO treatment. The first PCoA axis accounted for 60.44% of the total variability, as shown in Fig. 4C.

To demonstrate the compositional differences of rhizosphere soil bacterial communities between bulk ZnO and ZnO NPs treatment, a family-level analysis was conducted. Specifically, the treatment with ZnO NPs resulted in a decrease in the relative abundance of *Gemmatimonadaceae* from 10.12% to 8.2%, *Sphingomonadaceae* from 9.51% to 7.22%, SC

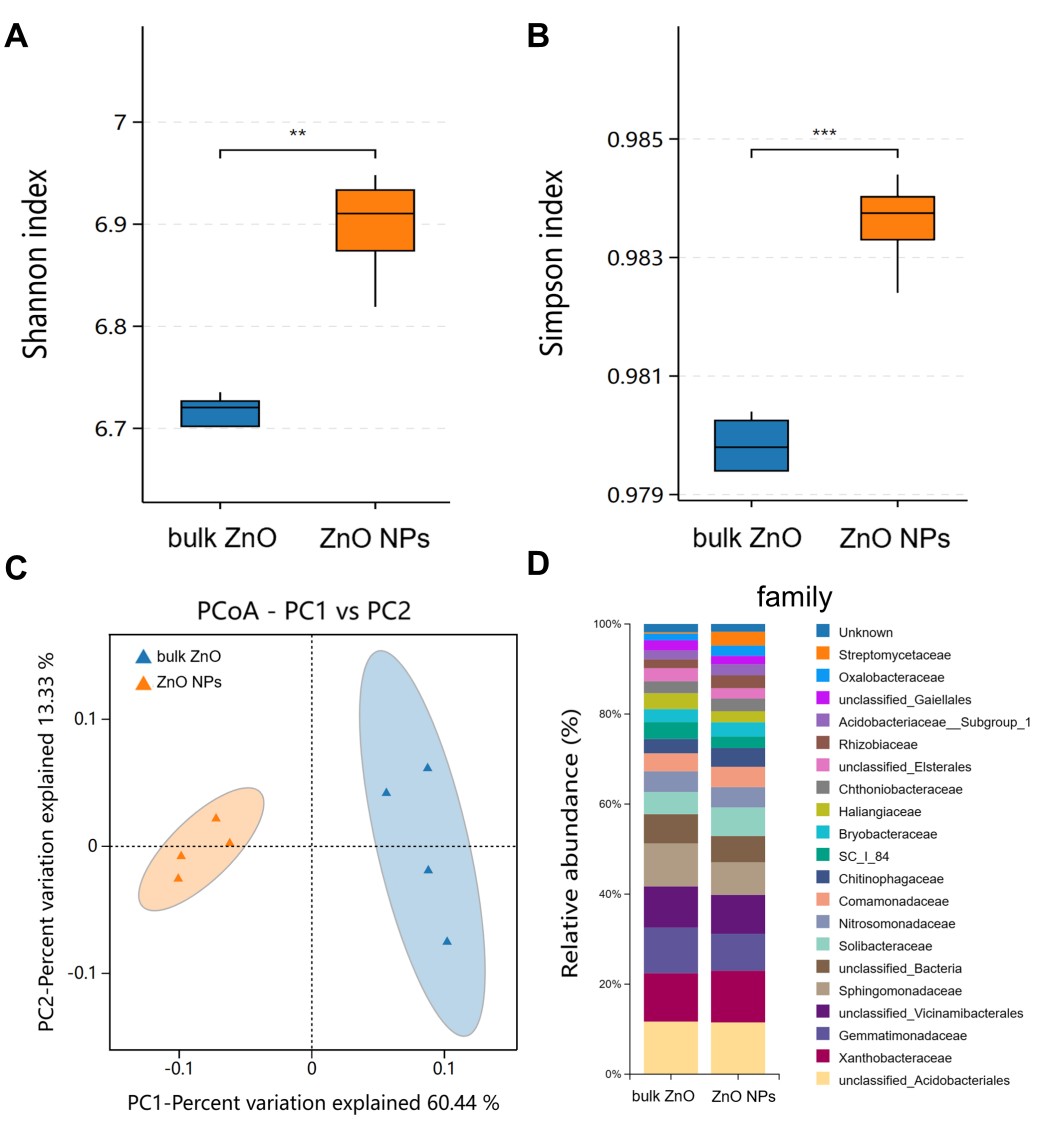

**Figure 4  Diversity and community structure of bacteria in rhizosphere soil under bulk ZnO and ZnO NPs treatments.** (A–B) Alpha diversity indexes (Shannon and Simpson). (C) PCoA based on Bray-Curtis dissimilarities. (D) Composition of bacterial communities at family level.

I 84 from 3.76% to 2.56%, *Haliangiaceae* from 3.52% to 2.44%, and *Unclassified_Elsterales* from 2.89% to 2.3%, among others. On the other hand, there was an increase in the levels of *Streptomycetaceae* from 0.29% to 3.06%, *Oxalobacteraceae* from 1.49% to 2.29%, *Rhizobiaceae* from 1.97% to 2.86%, *Chitinophagaceae* from 3.18% to 4.17%, and *Solibacteraceae* from 4.92% to 6.34% (Fig. 4D). Specifically, the treatment with ZnO NPs led to a reduction in the relative abundance of *SC I 84* showed a decrease from 3.76% to 2.56%, accounting for a reduction of 31.89%. *Haliangiaceae* decreased from 3.52% to 2.44%, resulting in a decrease of 30.42%. *Sphingomonadaceae* decreased from 9.51% to 7.22%, indicating a decrease of 24.07%. *Unclassified_Elsterales* decreased from 2.89% to 2.3%, representing a decrease of 20.21%. Additionally, *Gemmatimonadaceae*

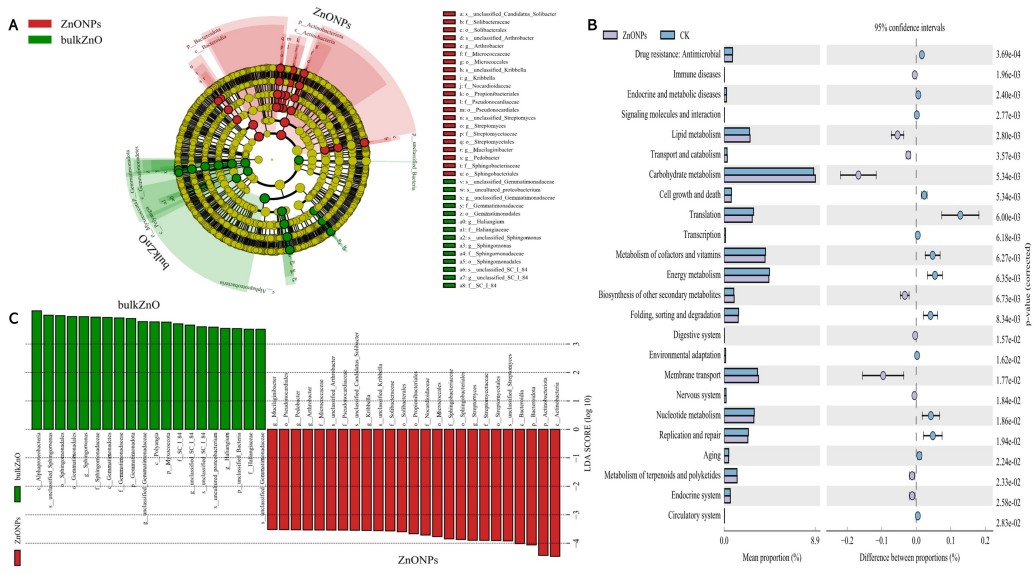

**Figure 5** **Identifcation of compositional diferences and relative abundance of predicted metabolic potential for bacterial communities in rhizosphere soil between bulk ZnO and ZnO NPs treatments.** (A) Taxonomic cladogram obtained from LEfSe (LDA scores (log10) > 3.5, $P < 0.05$). Abundance of taxa that differ between bulk ZnO (red bars) and ZnO NPs (green bars) treatments. (B) Mean of the relative abundance of predicted metabolic potential of rhizosphere bacteria using PICRUSt2. $P < 0.05$.

from 10.12% to 8.2%, showing a decrease of 18.92%. In contrast, there was an increase in the levels of *Streptomycetaceae* from 0.29% to 3.06%, representing an increase of 941.58%. *Oxalobacteraceae* increased from 1.49% to 2.29%, indicating an increase of 54.42%. *Rhizobiaceae* increased from 1.97% to 2.86%, showing an increase of 44.71%. Moreover, *Chitinophagaceae* increased from 3.18% to 4.17%, accounting for an increase of 30.9%, while *Solibacteraceae* increased from 4.92% to 6.34%, representing an increase of 28.82% (Fig. 4D). To identify the specific bacteria associated with the ZnO NPs treatment and distinguish the predominant taxa in the rhizosphere soil, LEfSe was utilized to generate a cladogram (all LDA scores (log10) >3.5, $P < 0.05$). The circles, representing the classification levels from phylum to species, were ordered from inside to outside. Each small filled circle indicated a classification at that level, with the size proportional to its relative abundance. The relative abundances of *Bacteroidota* and *Actinobacteriota* in the ZnO NPs treatment were highly significantly higher compared to the bulk ZnO treatment. Specifically, the relative abundance of Bacteroidota increased by 47.55%, while *Actinobacteriota* showed a 41.62% increase in the ZnO NPs treatment, as demonstrated in Fig. S2 ($P < 0.01$). These augmented levels of *Bacteroidota* and *Actinobacteriota* were considered as the biomarkers of the ZnO NPs treatment (Figs. 5A & 5C).

In order to explore the functional changes of rhizosphere bacteria under different treatments in this study, PICRUSt2 (Hierarchy level 2) was used to analyze the changes of the rhizosphere bacterial community of *A. carmichaelii*. Results based on KEGG database (Kyoto encyclopedia of genes and genomes) were indicated in Fig. 5B and Table S2. PICRUSt2 analysis revealed significant differences in the functional abundance of bacteria

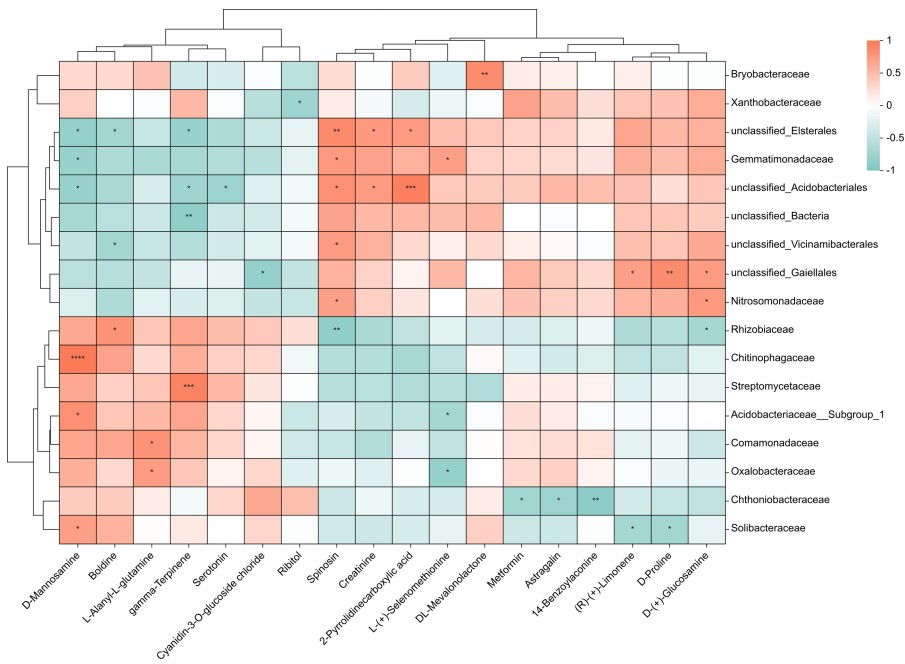

**Figure 6** **Integrated correlation analysis of rhizosphere bacteria and *A. carmichaelii* metabolites.** Asterisks indicate significant diferences between groups: **$P < 0.01$, *$P < 0.05$, Pearson test.

from bulk ZnO and ZnO NP treatments, including changes in 22 metabolic pathways, such as carbohydrate metabolism, energy metabolism, cofactor and vitamin metabolism, membrane transport, and nucleotide metabolism.

## Covariation between bacterial and metabolites of the lateral roots

To investigate the relationship between 38 upregulated/downregulated metabolites (Table S1 & Fig. 3B) and the top 20 most abundant bacteria at the family level, we conducted a correlation analysis. We obtained the covariation between rhizosphere soil bacteria and metabolites in the lateral roots of *A. carmichaelii* using Spearman's rank correlation with two-tailed nominal *p* values (Fig. 6). Only metabolites and bacteria displaying significant correlations were included in the figure.

The results presented in Fig. 6 showed significant differences between the bulk ZnO and ZnO NP treatments. The lateral roots of *A. carmichaelii* exhibited up-regulated expression of L-Alanyl-L-glutamine, 14-Benzoylaconine, Serotonin, D-Mannosamine, Metformin, and DL-Mevalonolactone in response to ZnO NPs treatment. What is intriguing is that their expression levels displayed positive correlations with specific microbial communities and reduced negative correlations. For example, the abundance of *Solibacteraceae* and *Chitinophagaceae*, which showed positive correlation with D-Mannosamine expression, significantly increased, while *Gemmatimonadaceae* and *unclassified_Elsterales*, which were negatively correlated with D-Mannosamine expression, decreased. Similarly, the expression levels of L-(+)-Selenomethionine, Ribitol, and (R)-(+)-Limonene were downregulated in the lateral roots of *A. carmichaelii* treated with ZnO NPs. Interestingly, bacteria that

negatively correlated with these metabolites were more abundant while those that positively correlated with them were less abundant.

## DISCUSSION

Soil trace elements, such as zinc (Zn), copper (Cu), manganese (Mn), iron (Fe), and boron (B), are essential for the growth and development of plants, but are only required in small quantities (*Jatav et al., 2020*). They contribute to the enhancement of plant growth through diverse mechanisms. These mechanisms include regulating key enzyme activity, influencing gene expression and transcription processes, as well as impacting the synthesis, transport, and metabolism of plant hormones (*Merchant, 2010*; *Lewis, 2019*). In this study, we evaluated the influence of trace elements on the growth and production of monoester-type alkaloids in *A. carmichaelii*. Our findings suggest that AMn in the soil is significantly and positively correlated with yield of the plant. Manganese is a crucial component of various metabolic enzymes, such as malic dehydrogenase and oxalosuccinic decarboxylase and is essential for plant growth and metabolic function (*Arif et al., 2016*). AZn in soil is significantly and positively correlated with yield and the content of monoester alkaloids in *A. carmichaelii*. Zinc, as a structural component of numerous proteins such as transcription factors and metalloenzymes, is one of the essential trace elements required for normal, healthy growth and reproduction in plants. If plants lack sufficient amounts of zinc, they may experience physiological stresses due to the failure of critical metabolic processes in which zinc plays an essential role (*Sadeghzadeh, 2013*). It is noteworthy, however, that TZn is negatively correlated with the content of monoester alkaloids. There is no significant correlation between TZn and AZn, indicating that blindly adding zinc may possibly reduce the content of monoester alkaloids. Our research also has revealed an antagonistic relationship between iron and zinc in soil, this phenomenon is attributed to the strong affinity of Fe oxide minerals present in soil, which tend to adsorb other metal cations, including Zn, onto their surfaces, ultimately reducing the availability of Zn to plants and resulting in Zn deficiency (*Vasu et al., 2021*). Regrettably, zinc fertilizers are often immobilized in soil and exhibit low effectiveness (*Guo et al., 2016*).

Overall, our results suggest that optimizing zinc availability can be an effective strategy for enhancing the growth and alkaloid content of *A. carmichaelii*. The application of traditional zinc fertilizers has encountered a bottleneck, and scholars are actively seeking new ways to improve the utilization of zinc fertilizers. Among them, ZnO NPs have entered people's field of vision. In this study, we examined the impact of ZnO NPs on the growth, metabolites, and monoester-type alkaloids of *A. carmichaelii*, as well as the rhizosphere microbiome.

Compared to distilled water and bulk ZnO treatments, the application of ZnO NPs led to significant improvements in plant height, fresh weight, and monoester-type alkaloid levels in *c*. The unique properties of ZnO NPs, including their small size, large surface area, solubility, and diffusion capacity, facilitate rapid absorption of released zinc by plants, thereby meeting their nutritional needs and promoting growth and development (*Bala, Kalia & Dhaliwal, 2019*). Moreover, ZnO NPs exhibit high mobility, meaning that
a small amount of fertilizer is sufficient to achieve the same effect as traditional fertilizers, resulting in cost savings (*Prasad et al., 2012*; *Zulfiqar et al., 2019*). Previously, *Salama et al. (2019)* found that spraying low concentration (0–40 mg/L) ZnO NP suspensions on *Phaseolus vulgaris* leaves, with a concentration of 30 mg/L led to the highest plant dry weight and yield. *Hyoscyamus reticulatus*, the main source of tropane alkaloids for treating Parkinson's disease, was found to have significantly increased expression of the hyoscyamine-6-beta-hydroxylase enzyme gene associated with the synthesis of tropane alkaloids and corresponding increase in tropane alkaloid content after treatment with 100 mg/L ZnO NPs (*Asl et al., 2019*).

Further investigation into the metabolic profile of lateral roots of *A. carmichaelii* revealed significant changes in metabolite content when comparing ZnO NPs and bulk ZnO treatments. Among the impacted pathways were arginine and proline metabolism, aminoacyl-tRNA biosynthesis, flavone and flavonol biosynthesis, glutathione metabolism, alkaloid biosynthesis, ascorbate and aldarate metabolism, tryptophan metabolism, and amino sugar and nucleotide sugar metabolism with 23 metabolites up-regulated and 15 metabolites down-regulated. The alterations in metabolite abundance within a tissue could indicate the inhibition or activation of specific metabolic pathways. Three of the perturbed biological pathways were related to amino acid metabolism. Amino acids play a critical role in various physiological processes of plants, serving as building blocks for proteins and osmolytes, regulating ion transport, and affecting the synthesis and activity of vital cellular enzymes (*Ghosh et al., 2022*). They also serve as signaling molecules and regulate stress-responsive genes (*Shah et al., 2021*). Therefore, changes in amino acid profiles could indicate a reconfiguration of nitrogen metabolism aimed at managing plant growth or development and modulating carbon and nitrogen status. This also explained the significant increase in height and fresh weight of lateral roots in *A. carmichaelii* under ZnO NPs treatment. Aminoacyl-tRNA transfers amino acids to the ribosome for protein synthesis, thus playing an essential role in translation in mitochondria, chloroplasts, and cytoplasm (*Ostersetzer-Biran & Klipcan, 2020*). The upregulation of Aminoacyl-tRNA biosynthesis indicates an increase in the rate of cellular metabolism and protein synthesis through various mechanisms. Alkaloids are a large cluster of molecules found in Mother Nature all over the world. They are all secondary compounds and collection of miscellaneous elements and biomolecules, derived from amino acids or from transamination (*Dey et al., 2020*). Arginine and proline metabolism, as well as tryptophan metabolism, can provide the necessary precursor molecules for alkaloid biosynthesis (*Yang et al., 2022*; *Zhou & Chen, 2022*). Therefore, there is a strong interconnection between arginine and proline metabolism, tryptophan metabolism, and the synthesis of alkaloids. The upregulation of pathways involved in alkaloid biosynthesis, arginine and proline metabolism, and tryptophan metabolism, resulted in increased levels of alkaloids in the lateral roots of *A. carmichaelii*.

Microorganisms play a vital role in soil ecosystems, and the application of ZnO NPs in this study significantly impacts the structure and functions of rhizosphere bacteria. For instance, LEfSe results indicated at the phylum level, under the ZnO NPs treatment, *Bacteroidota* exhibited a 47.55% relative abundance increase, while *Actinobacteriota* showed

a relative abundance increase of 41.62%. These findings demonstrate that both *Bacteroidota* and *Actinobacteriota* displayed significantly higher relative abundances in the ZnO NPs treatment compared to the bulk ZnO treatment. These phyla are involved in the nitrogen cycle and possess the potential to regulate soil nitrate concentrations (*Ma et al., 2023*). At the family level, *Streptomycetaceae* displayed the greatest increase in abundance under ZnO NPs treatment, rising from 0.29% to 3.06%, representing a remarkable increase of 941.58%. *Streptomycetaceae* plays a vital role in the decomposition of complex organic molecules, such as lignocellulose, cellulose, xylan, and lignin, which are indispensable for soil organic matter catabolism (*Olanrewaju & Babalola, 2019*). Based on the PICRUSt2 analysis, ZnO NPs treatment improved the functional category levels of rhizosphere soil bacteria. Specifically, the treatment resulted in significant improvements in membrane transports, which are essential for cell survival and can help to overcome environmental stresses (*Fei et al., 2020*). Additionally, the ZnO NPs treatment enhanced carbohydrate metabolism, promoting the turnover rate of available carbon by microbes (*Luo et al., 2020*). In this study, most of the bacterial communities with increased abundance showed a positive correlation with upregulated metabolite expression, while those with decreased abundance displayed a negative correlation. Conversely, the bacterial communities with decreased abundance were positively correlated with downregulated metabolite expression, while those with increased abundance showed a negative correlation. Previous studies have suggested that rhizosphere bacteria can stimulate plant growth through both direct and indirect mechanisms (*Arif, Batool & Schenk, 2020*; *Haskett, Tkacz & Poole, 2021*). The observed positive correlations between metabolites and species may suggest that rhizosphere bacteria promote the production of these metabolites.

## CONCLUSION

In this study, the use of ZnO nanoparticles significantly promoted the growth of *Aconitum carmichaelii* and increased the content of monoester-type alkaloids (mainly 14-Benzoylaconine), while also regulating the plant's metabolism and rhizosphere microbiome. These findings demonstrate the immense potential of ZnO NPs in promoting plant growth and enhancing medicinal properties, making them a promising alternative for sustainable agricultural practices.

### Funding

This work was financially supported by Sichuan Province Science and Technology Support Program (No. 2021YFN0119 & No. 2022JDRC0123), National College Students Innovation and Entrepreneurship training Project (No. 202214389051), the Program for Innovative Research Team of Chengdu Normal University (No. CSCXTD2020A04), and "14th Five-Year Plan" Breeding Key Project in Sichuan Province (No. 2021YFYZ0012). The funders had no role in study design, data collection and analysis, decision to publish, or preparation of the manuscript.

## Grant Disclosures

The following grant information was disclosed by the authors:

Sichuan Province Science and Technology Support Program: 2021YFN0119, 2022JDRC0123.

National College Students Innovation and Entrepreneurship: 202214389051.

Program for Innovative Research Team of Chengdu Normal University: CSCXTD2020A04.

Breeding Key Project in Sichuan Province: 2021YFYZ0012.

## Competing Interests

The authors declare there are no competing interests. Ye Tian and Yue-fei Geng are employed by Sichuan Jianengda Panxi Pharmaceutical Co. LTD

## Author Contributions

- Cun Chen conceived and designed the experiments, performed the experiments, analyzed the data, prepared figures and/or tables, authored or reviewed drafts of the article, and approved the final draft.
- Yu-yang Zhao performed the experiments, analyzed the data, prepared figures and/or tables, and approved the final draft.
- Duo Wang analyzed the data, prepared figures and/or tables, and approved the final draft.
- Ying-hong Ren performed the experiments, analyzed the data, prepared figures and/or tables, and approved the final draft.
- Hong-ling Liu analyzed the data, prepared figures and/or tables, and approved the final draft.
- Ye Tian performed the experiments, prepared figures and/or tables, and approved the final draft.
- Yue-fei Geng performed the experiments, prepared figures and/or tables, and approved the final draft.
- Ying-rui Tang performed the experiments, prepared figures and/or tables, and approved the final draft.
- Xing-fu Chen conceived and designed the experiments, performed the experiments, authored or reviewed drafts of the article, and approved the final draft.

## Data Availability

The data is available at NCBI GEO: PRJNA990327.

## Supplemental Information

Supplemental information for this article can be found online at http://dx.doi.org/10.7717/peerj.16177#supplemental-information.

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
