# Peer review of "Effects of nanoscale zinc oxide treatment on growth, rhizosphere microbiota, and metabolism of Aconitum carmichaelii"

_PeerJ, doi:10.7717/peerj.16177_

## Round 0.1 · original submission · Major Revisions

The authors need to include suggestions or comments given by each reviewer. Include all suggestions and then re-submit again.

Reviewer 1 ·

Basic reporting

The manuscripts writes in a clear and logical way but some small problems needs to be resolved. Examples below:

1. Line 31: Italic font for Bacteroidota, Actinobacteriota. Similarly, please double check all taxonomy name to make sure they are italic font.
2. Line 58 and 92: micro-fertilizers.
3. Line 224: 16S rRNA
4. Line 247, it should be and the root parts because there is no turning points.



Suggestions on introduction:
In 3rd paragraph, add 1-2 sentences about Mn since in figure 1 it also has high correlation with yield. Even only elements with highest correlation value is discussed, you should have 1/2 sentences to mention the general mechanisms of trace elements on yield. Moreover, what is the mechanisms of ZnO/Fe2O3 NPs on its yield? If no, please mention few recent advances studies the detailed mechanisms.

Suggestions on results.
1. Please include the p value for statistical analysis in the entire manuscript. For example, in section 3.1 line 233.The results show that the content of AMn in the soil is significantly and positively correlated with yield (p < ?).

Experimental design

The experimental design and setups are well organized.

Validity of the findings

Overall, this manuscript is analyzed in a logic, organized, and statistically tested manner. Just some minor problems that need revision.
1. in line 278-281: Fig 3C, please at least label what metabolites category in the red colors are corresponding to since the x/yticklabels are not informative? Similarly for blue sections.

2. in line 282-293: Fig 3D, please label the pathway in the figure.

3. Line 304, number 63.99% is not corresponding to the image label 60.44%.

4. Line 305-312: Do we have statistical analysis for Figure 4 to show the decrease is statistically significant? If not, the conclusion can not be reported.

5. In discussion section, there is no reports about how previous studies showing the mechanisms of trace elements on yield. I suggest authors to include one paragraph to discuss the potential mechanisms.

Reviewer 2 ·

Basic reporting

I need some explanation in this manuscript.

Experimental design

The experimental design was quite clear. The survey methodology is unclear. Please describe it in more detail.

Validity of the findings

no comment

Additional comments

Please check your methodology, result, discussion, and conclusion.

Annotated reviews are not available for download in order to protect the identity of reviewers who chose to remain anonymous.

Reviewer 3 ·

Basic reporting

Overall, this study is interesting to investigate the roles of trace elements in the medical plant- Aconitum carmichaelii growth. They discovered highly correlation between the soil zinc and plant yield and alkaloid content. Moreover, the authors identified some differentially expressed metabolites between treatments with ZnO and ZnO-NPs. I have several comments for this manuscript.

Experimental design

The design is sound.

Validity of the findings

The findings are interesting and will attract general interest.

Additional comments

1. Fig. 1, how do the authors determine the significance in Pearson's correlation analyses as they have one asterisk and two asterisks in the figures?
2. In Fig. 2, the pictures in A-C should have corresponding labels. Which one is CK, ZnO, or ZnO-NPs?
3. Fig. 3, in B, what is the cut-off p-value for the significant analyses? It is shown in Figure 3B that the cut-off is -log10(p-value)=1 and this indicates that the cut-off of the p-value is 0.1. However, p=0.05 or 0.01 is used for significance generally. If the authors used the p=0.1 as the cut-off, please showed the clear definition in the method section.
4. Also in Fig. 3B, what are the square and triangle in the right bottom corner? They might be removed.
5. Fig 3D, What are the numbers in the Bubble plot? Please indicate the descriptions of numbers in the legend.
6. The general writing is OK, however, it also might be improved with writing revision by native speakers.

---

## Round 0.2 · accepted · Accept

Authors included reviewers' suggestions. The manucript is good to accept.

Reviewer 1 ·

Basic reporting

The revised reporting is quite clear and unambiguous with professional writing and figures.

Experimental design

Well defined and is meaningful.

Validity of the findings

The conclusions are well stated and linked.

Reviewer 2 ·

Basic reporting

1. Please rephrase your manuscript. The similarity is more than 27% and must be <20%.
2. Please check your references after adding some citations

Experimental design

no comment

Validity of the findings

no comment

Additional comments

Please check the typo and format of this journal.

Annotated reviews are not available for download in order to protect the identity of reviewers who chose to remain anonymous.

Reviewer 3 ·

Basic reporting

The revision has answered my concerns, and I recommend accepting the current version. Thanks!

Experimental design

No comment

Validity of the findings

No comment

Additional comments

No comment